# The Russian war in Ukraine increased Ukrainian language use on social media

Daniel Racek [1✉], Brittany I. Davidson [2], Paul W. Thurner [3], Xiao Xiang Zhu [4] & Göran Kauermann[1]

The use of language is innately political, often a vehicle of cultural identity and the basis for nation building. Here, we examine language choice and tweeting activity of Ukrainian citizens based on 4,453,341 geo-tagged tweets from 62,712 users before and during the Russian war in Ukraine, from January 2020 to October 2022. Using statistical models, we disentangle sample effects, arising from the in- and outflux of users on Twitter (now X), from behavioural effects, arising from behavioural changes of the users. We observe a steady shift from the Russian language towards Ukrainian already before the war, which drastically speeds up with its outbreak. We attribute these shifts in large part to users' behavioural changes. Notably, our analysis shows that more than half of the Russian-tweeting users switch towards Ukrainian with the Russian invasion. We interpret these findings as users' conscious choice towards a more Ukrainian (online) identity and self-definition of being Ukrainian.

[1] Institute of Statistics, Ludwig-Maximilians-University Munich, Munich, Germany. [2] School of Management, University of Bath, Bath, UK. [3] Institute of Political Science, Ludwig-Maximilians-University Munich, Munich, Germany. [4] School of Engineering and Design, Technical University of Munich, Munich, Germany. ✉email: daniel.racek@stat.uni-muenchen.de

Social media is critically important in today's society[1–3]. In recent years, it has played a key role in a number of political shifts and crises[4,5]. While social media has been found to amplify all manners of misinformation, propaganda, populism, and xenophobia[6–8], it can also serve as a mechanism to call for aid and as a source for live updates of major events unfolding[9–12].

In this article, we analyse language use of Ukrainian citizens on social media before and during the Russian invasion of Ukraine (subsequently referred to as war), where after years of tensions and open aggression between Russia and Ukraine[13], on 24th February 2022, Russian forces began to invade and occupy parts of Ukraine[14]. At the time of writing, it has been estimated that the war has led to over 23,000 civilian casualties[15] and hundreds of billions of dollars worth of damage[16,17]. This has caused world-wide unrest, alongside 8.2 million Ukrainian refugees recorded across Europe and 5 million registered for temporary protection[18,19].

The war in Ukraine is also taking place in the digital era, with social media coverage documenting the horrific events in up to real-time. This provides a unique digital trace of many first-hand accounts of the war, as citizens are communicating among each other and to the public. This is generally known as crisis informatics, whereby social media data are utilized before, during, or after emergency events for use cases such as disaster monitoring, management, and prevention[9,12,20–22]. Recent studies have demonstrated that tweets can capture events of political violence[23] and can help in monitoring and understanding intra-country conflicts[24].

In our work, the user's choice of language in a tweet is of particular interest. Many people across the world (including most Ukrainian citizens with the Russian and Ukrainian language[25]) are multilingual. This multilingualism comes with a number of links to an individual's identity, as someone may speak one language at work, but another one at home with their family. Thus, different languages are spanning across multiple facets of one's identity[26]. These context-based adaptations of our self-presentation and behaviour are expected by those around us[27–30]. Hence, it is important to note that a user's choice of language online can be argued as an active choice to communicate and a way they seek to present themselves to their audience[26]. For example, many non-natives switch to English in order to ensure a wider intelligibility online[31].

The use of language is also inherently political. Languages can be the cause of conflict and they are often incorporated in cultural and ethnic identity definition and are the basis for nation building and political change[32,33]. After the dissolution of the USSR, most post-soviet countries introduced new language laws in order to assert their original native language and build a new nation[32,34]. In Ukraine, after their independence, many people were considering themselves Russians by nationality or Ukrainian with Russian as their main native language[35,36]. While the government aimed to reverse those effects, they were only moderately successful in achieving this goal, as census results show[35–37]. Only more recently, with the Euromaidan protests and the Russian military intervention in Crimea and the Donbas, surveys between 2012 and 2017 show a consistent and substantial shift away from Russian ethnic and linguistic identification towards Ukrainian practice[37].

We investigate language choice and tweeting activity on Ukrainian Twitter (now called X) from January 2020 to November 2022 using over 4 million geo-tagged tweets from more than 62,000 different users. In doing this, we study how Ukrainian citizens (and non-citizens living there) respond to their country being aggressively attacked and invaded by its direct neighbour they share a long history and language with, and how the use of language evolved before and during this war. Our study

allows us to follow the same set of users and observe their (change in) behaviour over both the short- and longer-term as the war breaks out and continues to unfold on an individual level. Hence, we are able to comment on recent news articles outlining shifts in language use from Russian to Ukrainian as a direct result of the war[38,39]. Moreover, we are able to monitor long-term language trends even before the war without the necessity of relying on small-scale surveys nor the infrequent censuses, the last one of which was conducted in 2001.

More specifically, we study overall trends in the number of tweets in the three main languages (Ukrainian, Russian, English) over time. Second, we investigate how these trends translate to users' individual tweeting activity and if changes result from the in- and outflux of users, common in online communities[40–42], or if they result from users changing their behaviour over time[43–45]. We quantify the magnitude of both effects respectively. Third, we study if changes in users' tweeting activity originate from shifts between languages and quantify the magnitude of these shifts. Fourth and finally, we take a closer look at those users that switch from predominately tweeting in Russian to predominately tweeting in Ukrainian with the outbreak of the war.

## Methods
This study was ethically approved by the ethics commission of the faculty of mathematics, computer science and statistics at Ludwig-Maximilians-Universität (LMU) München, Germany. The reference identifier is EK-MIS-2022-127. We did not pre-register this study. No information on user demographics such as age, sex, gender or race were collected or determined and - in accordance with the ethics commission - no informed consent by the Twitter users was obtained.

### Data
*Data collection & final dataset.* We collected tweets from 9th January 2020 to 12th October 2022 using the 1% real-time stream of the Twitter API. During collection, we filtered the data such that we only gathered tweets containing geo-information from the API. We then manually filtered the dataset to only retain tweets from Ukraine (denoted by the "UA" country tag), as common in the literature[46], and excluded any retweets, which left us with primary tweets, quotes and replies, all of which contain original tweet texts.

This dataset obtained from the 1% stream consisted of 4,102,982 tweets. As we began cleaning, we noticed gaps with missing tweets, most likely due to server and internet outages during the real-time data collection process. Hence, we retrospectively identified and filled all gaps. To do this, we first identified all time windows >10 min without any tweet and added them to our download queue. Days with more than two of such time windows were added to the queue as a whole. We then queried the Twitter Research API 2.0 using the *tweets/search/all* endpoint to obtain tweets with Ukrainian geoinformation for all time windows in this queue and added the newly obtained tweets to our original dataset. Finally, we repeated this process for the 15 days with the least amount of tweets in our dataset. After removing all duplicates, this meant we added a total of 350,359 additional tweets to our dataset this way. Our subsequently conducted sensitivity analysis shows that through the two-stage filtering process combined with the recollection efforts, we were able to recover almost all geo-tagged tweets from Ukraine during this time period (see section "Sensitivity Analysis" for more info).

We conducted an extensive spam filtering scheme, in which we (1) removed any duplicate tweets, (2) identified and removed potential spam bots by training a bot detection model following[47], (3) removed users with >100 tweets per day, (4) only kept tweets

coming from official Twitter clients or Instagram, and (5) applied additional filtering rules specific to our dataset. This reduced our dataset from originally 4,453,341 tweets (62,712 users) down to 2,845,670 tweets (41,696 users). For a more extensive description and rationale see section "Data Cleaning".

*User characteristics.* Unsurprisingly, social media is popular in Ukraine, particularly among the younger generation, with almost all citizens aged 18-39 in 2021 reporting that they use social media. For Twitter, user statistics are as follows: 18–29 (13% usage), 30–39 (8%), 40–49 (7%), 50+ (1%)[48].

We provide an overview and descriptive statistics on all user attributes as available from the API in Supplementary Table 4. The relevant user attributes for our main results and their assigned names are described in the following. Followers are the number of accounts that follow a user. Followings reports the number of accounts a user is following. The account age the number of months a user account has existed from account creation to their latest tweet in our dataset. The tweet frequency the number of tweets per day. The like frequency the number of liked tweets (by the user) per day. # of Tweets in Ukraine reports the total number of tweets in our dataset. All Twitter user attributes are a snapshot from the last time we observe a user's respective tweet in our sample.

As described in Supplementary Notes 1, we conduct multilingual topic modelling using BERTopic[49]. War topic 1 reports the number of tweets assigned to first war topic cluster (topic #1), which covers updates about the war and calls for help. War topic 1 (rel.) the relative share of tweets assigned to this topic. War topic 2 reports the number of tweets assigned to second war topic cluster (topic #3), which covers a more political side of the overall conflict. War topic 2 (rel.) the relative share. A full list of all topic clusters is available in Supplementary Table 1.

*Sensitivity analysis.* After data collection (before the cleaning), we evaluated the completeness of the dataset, i.e. whether we were able to recover most of the tweets published in Ukraine over the course of the study period, using the following strategy. We draw a random subset of 29 days from our analysis period and draw tweets from the Twitter Research API 2.0 using the *tweets/search/all* endpoint, which returns all historic tweets that have not been deleted since. We find a coverage of 98.24% (SD: 3.09%). More importantly, in the opposite direction we are only able to report a coverage of 77.67% (SD: 9.55%). Hence, employing our strategy using the real-time stream offers substantially more tweets, which have been deleted since (for more information on tweet deletion and its effects see ref. 50). Moreover, this suggests we were able to recover most of the geo-tagged tweets from Ukraine.

*Data cleaning.* For cleaning our dataset, we first train a Twitter bot detection model using a random forest (RF), as described in ref. 47. We use the exact same model as described in the authors' work (except for removing the attribute *profile_use_background_image*, which is no longer available from the Twitter API), using the training datasets *botometer-feedback*, *celebrity*, *political-bots*, as well as 100 manually labelled Twitter accounts from our dataset. To evaluate performance, we first set up a nested cross validation (CV) routine, with both a fivefold CV in the inner and outer loop. The inner CV is used for hyperparameter tuning, tuning both the number of trees as well as the minimum node size of the RF, whereas the outer loop is used for evaluating model performance. This results in an average area under the receiecer operator characteristics curve (AUROC) of 0.9837 and an average area under the precision-recall curve (AUPRC) of 0.7707. For our final model, we replicate this procedure, by setting up a 5-fold CV on the entire dataset to find the

best performing hyperparameters. We then train our RF on the entire dataset and use this model to identify bots and spam accounts in our dataset.

As we are only interested in removing the most prevalent spam, we opt for a conservative removal strategy to not falsely remove too many real and non-spam users. Hence, we only remove users with a predicted bot probability >50% and more than 10 tweets since account creation as well as users with a predicted bot probability >30% and more than 10,000 tweets. While thresholds of 50% and 30% respectively might not seem conservative, in the given setting, in which the bot class is heavily underrepresented (3.7% of observations in training dataset), an F1-optimizing threshold on the training dataset would lie far below that. We are somewhat less conservative with users that published over 10000 tweets, as in most cases they are spam accounts (e.g. related to bitcoins or NFTs). We do to not remove users with less than 11 tweets, as even for a human it becomes incredibly difficult to determine if a user is a bot with such limited amount of information to draw from. At the same time, we noticed a large influx of new users after the outbreak of the war who exclusively called for help in a short span of time, a behaviour which can easily be mistaken for a bot. Notably, we do not tune the optimal classification threshold, as the outbreak of the war in Ukraine represents an unprecedented event, with an unusual amount of new users joining (see Section "User Activity"). Hence, we expect the distribution between the target label (bot or human) and our features to be different between the bot training dataset and our Ukrainian dataset. Unfortunately, an extensive manual labelling strategy and more elaborate bot detection is beyond the scope of this work and would warrant its own paper. In summary, with this strategy we remove a total of 2021 users and their tweets from our dataset.

To further identify and remove potential spam accounts, we identify all accounts with more than 100 tweets on a single day (the mean is ~ 4.4 and the median = 2), and remove those 257 users from the dataset. We also noticed an unusual amount of Tweets containing the word "BTS" (45,579; referring to the Korean K-Pop band[51]) with spikes on specific days, which we subsequently filter out. Next, we identify and remove any tweets published by the same user that contain the exact same text as their previous tweet if both tweets were published within a one minute window. Fifth and finally, we filter out any tweets with the *source* attribute not being equal to Instagram or Twitter. That way, we discard any tweets automatically published by social media schedulers such as dlvr, which are often used by news agencies or other companies.

## Statistical modelling

*Tweet modelling.* We define the number of tweets $Y_{t,u,l}$ made in week $t$ by user $u$ in language $l$. As tweets are count data, we model the $Y_{t,u,l}$ to follow a Poisson distribution with intensity $\lambda_{t,u,l}$, where

$$\lambda_{t,u,l} = \exp(\mu + s_l(t) + W_{u,l}). \tag{1}$$

Here, $\mu$ is a general time-constant intercept, which captures the average tweet intensity over all users, languages and weeks. The $W_{u,l}$ are language-specific time-constant random intercepts for each user $u$, assumed to be normally distributed. They capture by how much the average tweeting behaviour (more or less tweets) of each user in each language differs from the general mean $\mu$. Finally, $s_l(t)$ denotes a smooth global time trend for each language $l$ (Ukrainian, Russian, English) and captures changes in the tweeting behaviour over all users over time. Hence, with the latter, we can measure behavioural changes of the users over time (e.g. are users tweeting more with the outbreak of the war?),

whereas the random intercepts measure changes in the user sample over time (e.g. are users that enter the platform after the war tweeting more on average?). This results in a generalized additive mixel model (GAMM). For more information, we refer the reader to ref. 52 and ref. 53. We fit the model with the R package *mgcv* v1.8.41[54] using the GAM implementation for very large datasets *bam*. To speed up the estimation, we use the discrete option, which discretizes covariates to ease storage and increase efficiency. For fitting $s_l(t)$, we employ thin plate regression splines. Our estimation sample consists of $y = 1,045,245$ observations, with $t = 143$ weeks, $l = 3$ languages and $u = 13,643$ users. For our fitted model, we report an explained deviance of 71.3%.

The effect sizes in the results are calculated as follows. For the behavioural effects we derive the change in $s_l(t)$ between two respective dates $t_1$ and $t_2$ and take the exp(.), i.e. $\exp(s_l(t_2) - s_l(t_1))$ for each language $l$. The result is the change in expected tweeting activity due to behavioural changes, when controlling for the in- and outflux of users. The sample effects are derived by averaging the random effects of the active users at the two respective dates and taking the exp(.), i.e. $\exp(\overline{W}_{t_2,l} - \overline{W}_{t_1,l})$. We define $\overline{W}_{t,l}$ as the average random effect in language $l$ over all users $u$ active at time point $t$. This captures the averaged change in expected tweeting activity due to a change in average tweeting intensity of the active users, when controlling for behavioural changes.

*Language modelling.* To model users' pairwise language probability, we refrain from a multinomial modelling strategy, as even with a weekly setup our dataset is particularly large. (To the best of our knowledge, a package with a parallel estimation routine for large datasets that can fit a GAMM for a multinomial distribution does not exist.) Instead, we model each pairwise probability separately through a binomial distribution. Our pairwise evaluation gives us a total of three different language pairs (UA over RU, UA over EN, RU over EN), for which we model the probability $\pi$ to tweet in language one (subsequently $l_1$) over language two (subsequently $l_2$). The order in which we specify these pairs is irrelevant, as the probability to tweet in $l_2$ over $l_1$ is simply $1 - \pi$. More specifically, we define $X_{t,u}$ as the number of tweets made in week $t$ by user $u$ in $l_1$. We assume $X_{t,u} \sim Binomial(n_{t,u}, \pi_{t,u})$, where $n_{t,u}$ denotes the total number of tweets made by user $u$ in week $t$ (sum of tweets in $l_1$ and $l_2$) and $\pi_{t,u}$ corresponds to the probability to tweet in $l_1$ over $l_2$. We assume that $n_{t,u}$ is known and instead model $\pi_{t,u}$ by setting

$$\pi_{t,u} = f(\mu + s(t) + W_u), \qquad (2)$$

where $f(.)$ is defined as the logistic function. Similarly to before, $\mu$ is a general time-constant intercept, which captures the average mean probability over all users and weeks to tweet in $l_1$ over $l_2$. Again, the $W_u$ are time-constant random intercepts for each user $u$ that capture by how much the average probability differs from the general mean $\mu$, and are assumed to be normally distributed. The smooth global time trend $s(t)$ captures changes in the probability over all users over time. Hence, as before, we can measure behavioural changes of the users over time with the latter (are users actively changing the language they are tweeting in?), whereas the random intercepts measure changes in the sample over time (how does the language probability of users entering/ leaving the platform evolve?). We estimate this model specification for all three aforementioned language-pairs with the R package *mgcv* v1.8.41[54] using the GAM implementation for very large datasets *bam*. To speed up the estimation, we use the discrete option, which discretizes covariates to ease storage and increase efficiency. For fitting $s(t)$, we employ thin plate regression splines. Users not tweeting in either of the two languages of the respective language pair, need to be discarded by definition. Hence, for UA or RU our estimation sample consists of of $x = 194,178$ observations, with $t = 143$ weeks and $u = 10,531$ users. For UA over EN: $x = 146,984$, $t = 143$, $u = 9,133$. For RU over EN: $x = 170,853$, $t = 143$, $u = 10777$. For our fitted models, we report explained deviances of: 85.8% (UA over RU), 90.5% (UA over EN) and 90% (RU over EN).

The coefficients of a logistic regression, as employed here, must be interpreted with respect to changes in the odds (also known as odds ratio). The odds ratio is defined as $odds = p/(1 - p)$. Hence, it describes how likely an event is going to happen compared to not happen. In this setting, it describes how likely it is to tweet in language 1 over language 2.

The effect sizes in the results are calculated as follows. For the behavioural effects we derive the change in $s(t)$ between two respective dates $t_1$ and $t_2$ and take the exp(.), i.e. $\exp(s(t_2) - s(t_1))$ for each of the three models. The result is the change in odds to tweet in $l_1$ over $l_2$ due to behavioural changes, when controlling for the in- and outflux of users. The sample effects are derived by averaging the random effects of the active users at the two respective dates and taking the exp(.), i.e. $\exp(\overline{W}_{t_2} - \overline{W}_{t_1})$ for each of the three models. We define $\overline{W}_t$ as the average random effect over all users $u$ active at time point $t$. This captures the averaged change in odds due to a change in average tweeting probability of the active users, when controlling for behavioural changes.

**Reporting summary**. Further information on research design is available in the Nature Portfolio Reporting Summary linked to this article.

## Results

**Descriptive findings**. To determine the language of a tweet, in accordance with the literature[55,56], we utilize the language field provided by the Twitter API. Ukrainian (35.8%) and Russian (35.4%) tweets are most prevalent in our dataset, followed by English (11.5%). A large proportion of tweets (11.1%) is labelled as "undefined", which among others consists of tweets that are too short, contain only hashtags, or only have media links. All other languages have shares of 1.2% or less. For the subsequent analysis we focus on tweets coming from the three main languages (English, Russian, Ukrainian) and discard all remaining tweets. A full breakdown of the language distribution is reported in Supplementary Figure 6.

In our dataset, there are clear trends in the aggregate over time (Fig. 1). In the beginning of 2020, we can see that Russian is the predominant language being used on Twitter in Ukraine, however, over time, this number gradually declines. The number of Ukrainian and English tweets on the other hand remains more or less constant over this initial time period. In the figure, we mark two key dates. On 11th November 2021, the United States officially report a mobilization of Russian troops along the Ukrainian border for the first time[57–59]. We will subsequently call this the first signs of aggression. 24th February 2022 marks the begin of the Russian invasion of Ukraine (subsequently referred to as outbreak of the war). As we approach this outbreak, there is a clear spike in tweets across all three languages, with a larger spike in both English and Ukrainian. Afterwards, English and Russian remain mostly constant, although the former on a much higher level than before. For Ukrainian, there is a clear upward trend in the daily number of tweets after the outbreak of the war.

Given these remarkable shifts in the number of tweets in the three considered languages, we want to investigate the underlying factors contributing to these changes. Note, that from the aggregate trends, we can not distinguish whether the observed

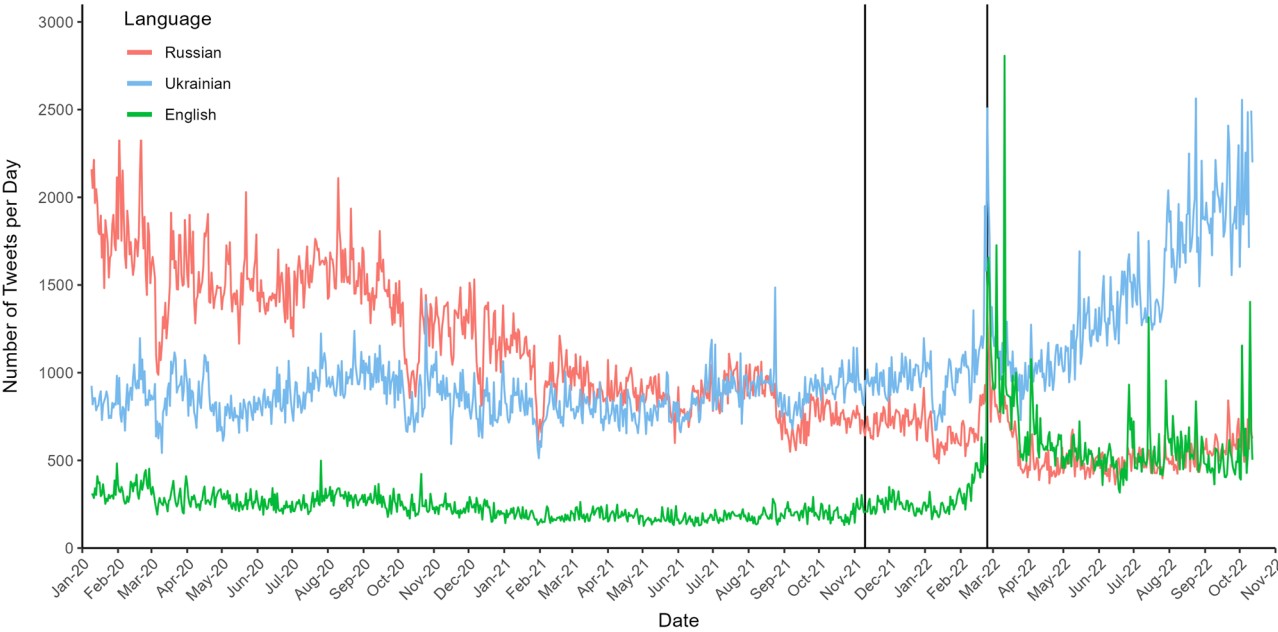

**Fig. 1 Daily number of tweets in the three most common languages.** Russian in red, Ukrainian in blue, English in green. From 9th January to 12th October (1008 days). The first vertical line denotes the mobilization of the Russian troops along the Ukrainian border (11th November 2021). The second line denotes the outbreak of the war (24th February 2022).

patterns are due to large in- and outfluxes of users, i.e. user turnover, which are common in online communities[40–42], or whether the actively tweeting users change their behaviour over time[43–45]. The disentanglement of this question is the aim of the rest of this article.

**User activity**. In order to address this question, we restructure our dataset by aggregating the number of tweets made by each user in English (EN), Ukrainian (UA), and Russian (RU) in each week. (Note, that we employ the Ukrainian country code "UA" instead of the official Ukrainian language tag "UK" in order to avoid confusion.) This allows us to study users' individual behaviour over time. To obtain reliable results, we restrict the further analysis to users who have tweeted in total at least ten times in any of the three languages. Furthermore, we choose weeks instead of days, as we are interested in general shifts and overall changes in behaviour over time, which are captured sufficiently well on a weekly basis. Through this weekly definition, we can dramatically reduce the size of our dataset, hence more complex modelling approaches become computationally feasible. We drop the first and last week in our dataset as these are incomplete (less than 7 days) and aggregate the remaining tweets on a weekly basis for each user and language. Finally within this, we are only considering weeks in which users are active (we define this as any week in which a user is tweeting at least once, as well as up to two weeks after), in order to account for the times in which users may be inactive for several weeks at a time or abandon their accounts. Thus, our new sample ranges from 13th January 2020 to 10th October 2022 and consists of 143 analysis weeks, 13,643 users and 1,045,245 observations.

Using this definition of user activity, we can visualize the total amount of active users as well as turnover rates (switch from active to inactive and vice versa) over time (Fig. 2). In the beginning of 2020, we have around 2800 active users per week. This number gradually decreases to roughly 1,800 until we approach the outbreak of the war. Afterwards, the number of active users starts increasing again. Note the drop and subsequent spike in activity shortly before and with the outbreak of the war.

Looking at the turnover rates, we find that there is a constant stream of ~250 (potentially different) users per week that switch from active to inactive and vice versa. The aforementioned spikes are also evident in these turnover rates. Finally, we find that there are roughly 50 users per week that join our sample for the first time and about the same amount that leave it altogether. Both of these numbers almost double after the outbreak of the war.

**Tweeting activity**. To obtain a better understanding on how the average active Ukrainian Twitter user changes over time, we visualize the average number of published tweets by a user in each language in Fig. 3a. From the figure, we can clearly see that there are substantial shifts. Overall, the average number of RU tweets per user decreases constantly over time (from over 6.5 to 2.1), the outbreak of the war being no exception. The average number of EN tweets decreases slightly until the war, where we notice a sudden uptick (from 0.7 to 2.8), followed by a steady decline. Meanwhile, the number of UA tweets slowly but steadily rises (from 2.4 to 2.9), with steeper increases after the first signs of aggression in November 2021 and no appearance of slowing down (5.3 at the end).

By combining these findings with Fig. 2, we can at least partially explain the aggregate trends evident in Fig. 1. While the active user sample is shrinking over time, those users that stay (and join) the sample are tweeting more in UA. Hence, there is no decrease in the overall amount of UA tweets. We find the exact opposite for RU tweets. As the number of active users is declining, the users that stay active are tweeting less in RU, resulting in the visible decrease of aggregate RU tweets over time. Notably, so far, we do not know, if those changes in the average amount of tweets per user are simply driven by shifts in our active user sample (i.e., are those users that initially tweet a lot in RU leaving over time and this is why we see this decrease in the average?), or, if these changes are (at least partially) driven by behavioural changes in those users that remain active on Twitter (i.e., are the same users tweeting less in RU over time?).

We address this through our tweet model described in Section "Tweet Modelling". We fit a generalized additive mixed model

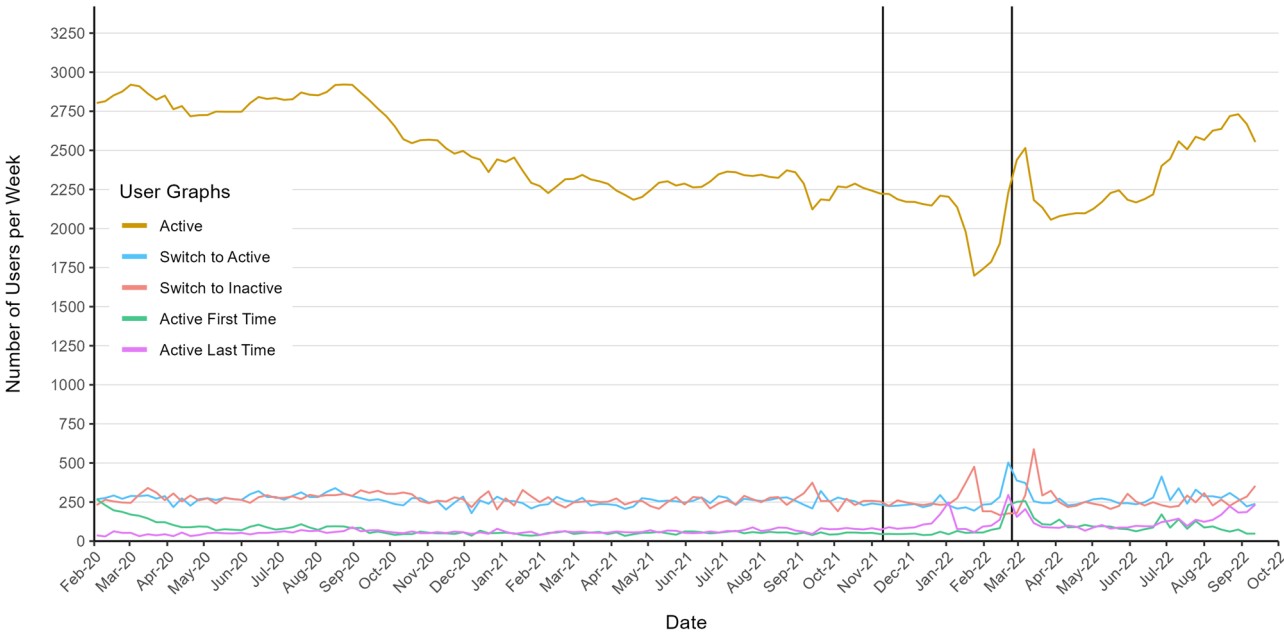

**Fig. 2 Weekly user activity graphs.** The brown graph reports the number of active users in each week. The blue (red) graph reports the number of users who switch to active (inactive), the green the number of users who switch to active for the first time, the purple the number of users who were active for the last time, i.e. drop out of the sample altogether. All graphs, but particularly the latter two, are skewed upwards respectively downwards towards beginning and end of the study period due to the nature of how the dataset is constructed. Hence, we drop the first and last three weeks for visualization purposes (137 total weeks left). The full plot is available in Supplementary Notes 5.

(GAMM) to predict the number of tweets made by each user in each language in each week, assuming a Poisson distribution. By incorporating both a smooth global time trend for each language, as well as user-specific random effects for each of the languages, we disentangle sample shifts (random effects) from behavioural changes (global trend). Hence, the former capture any changes in the population of active Ukraine-based Twitter users, while the latter strictly measures how these active users change their behaviour.

Figure 3b visualizes the average fitted sample (population) effects, i.e. the graphs depict how the average time-constant tweeting intensity in our active user sample changes over time due to user turnover. The figure shows, that the average RU tweeting intensity is mostly constant over time until November 2021, where aggression starts. From that point onward, in the span of only a few months, we see a decline of 21% in RU tweets from November 2021 to October 2022 (end of study period), solely attributed to changes in the user sample during that period. For EN, we find somewhat of an opposite effect. Similarly, there are only minor fluctuations until November 2021. But afterwards, there is a sharp increase of 107%. Taking a look at UA, we find a long-term increase of about 43% before the aggression starts. This increase comes to a hold shortly before the war, and considerably speeds up in the weeks after (+87%). All (relative) effect sizes calculated between the most relevant dates in our analysis period (start of study period, first signs of aggression, outbreak of war, end of study period) are reported in Table 1. We elaborate on this in Supplementary Notes 6, where we provide an additional figure, which illustrates sample changes over four-weekly intervals (Supplementary Fig. 9). From there, we can observe that the largest shifts clearly take place with and after the outbreak of the war. We also provide an alternative to Table 1, which measures the speed of change between the key dates in Supplementary Table 5. A full breakdown of all model coefficients is available in Supplementary Table 7.

Next, we will investigate behavioural changes using Fig. 3c. The graphs depict how the tweeting behaviour of the active users changes throughout the study period, when controlling for the user turnover (sample effects). Starting with RU, we notice that users are tweeting less and less over time. From January 2020 to November 2021, users tweet 49% less in RU due to behavioural changes. Subsequently, we see a small rise with the outbreak of the war (+5%), followed up by an even steeper decline (−24%). In contrast, UA is reasonably consistent in its use up until the start of aggression. From there, we observe a surge (+36%) until the outbreak of the war, followed by a gentler increase (+15%) after. Finally, looking more closely at EN tweeting behaviour, we can observe a general downward trend (−34%) until November 2021. Once the aggression starts, there is a huge spike (+130%), as users are tweeting a lot more in EN. After the outbreak of the war, this somewhat reverses (−40%), however, without dropping back down to pre-aggression levels. A full breakdown of all changes is reported in Table 1. Again, we elaborate in in Supplementary Notes 6. Supplementary Figure 9 shows that the largest behavioural shifts take place shortly before, with, and after the outbreak of the war. As a robustness check, we also pursue two alternative modelling strategies, one using factor smooths instead of random intercepts, the other implementing a regression discontinuity design, which are discussed extensively in Supplementary Notes 2.1 and 2.2 respectively. Both confirm the behavioural patterns described here.

Overall, we can conclude that there are only minor sample shifts pre-dating aggression that affect tweeting activity, but major shifts thereafter. In terms of behaviour, we can already observe steady changes early on, which considerably intensify with the war. However, as of yet, we cannot exactly pinpoint where those changes come from. Are users that already tweet in UA simply tweeting more with the outbreak of the war, or is it possible that users are actively switching the language they are tweeting in?

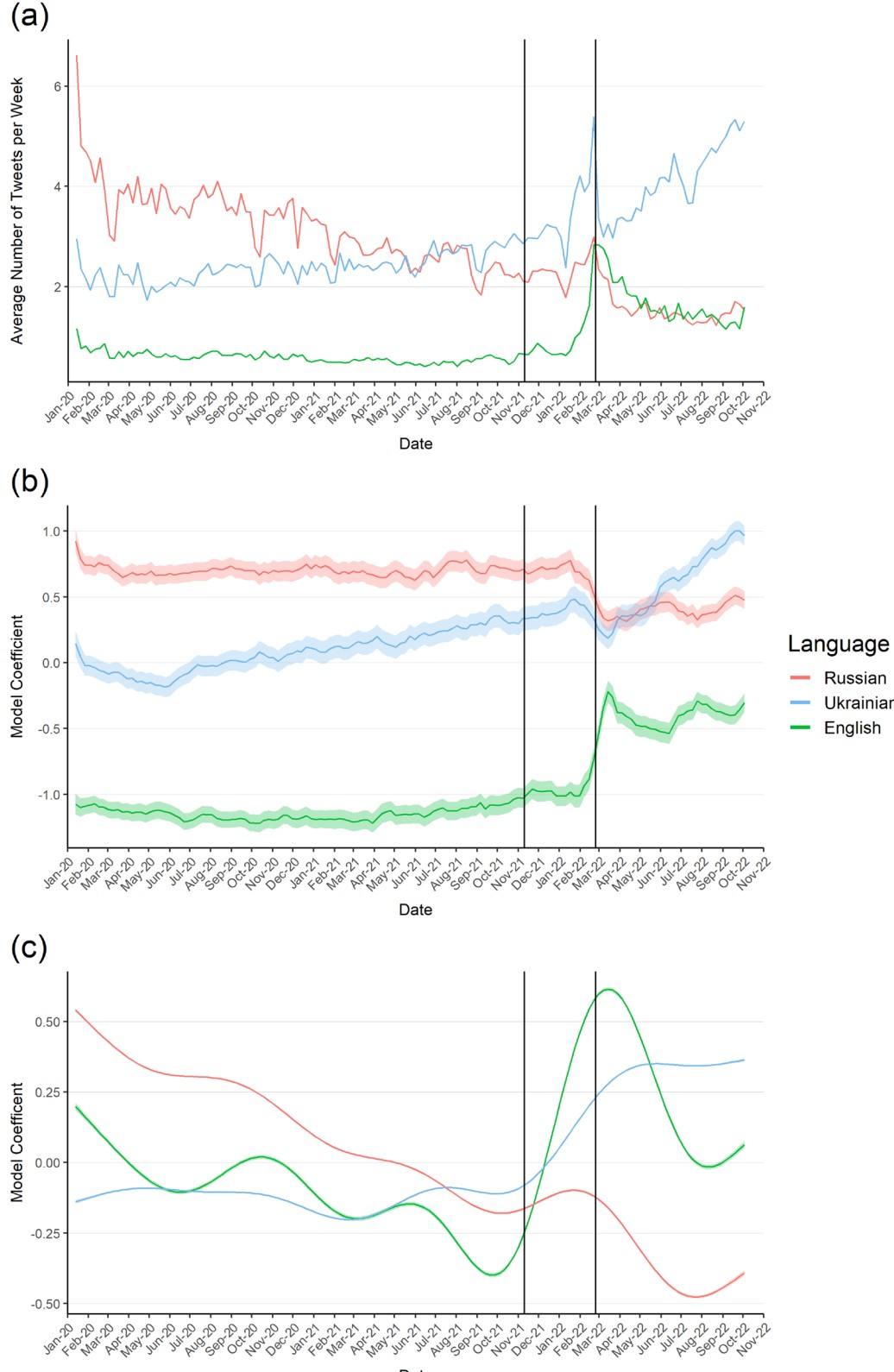

**Fig. 3 Changes in the number of tweets per user.** Russian in red, Ukrainian in blue, English in green. (**a**) visualizes the descriptive average number of tweets over time, (**b**) the sample effects (average random effects), (**c**) the behavioural effects (global trend). The shaded area depicts the 95% confidence interval. The first vertical line denotes the mobilization of the Russian troops along the Ukrainian border. The second line denotes the outbreak of the war.

**Table 1 Tweet activity effect sizes between key dates**

| Language | Sample effects | | | |
|---|---|---|---|---|
| | Start—Aggression | Aggression—War | War—End Study | Aggression—End Study |
| English | +6.16% | +34.87% | +53.14% | +106.54% |
| Ukrainian | +43.12% | −0.44% | +87.70% | +86.87% |
| Russian | −2.91% | −17.41% | −4.12% | −20.82% |
| | **Behavioural Effects** | | | |
| English | −34.41% | +130.11% | −39.98% | +38.09% |
| Ukrainian | +4.67% | +35.72% | +15.184% | +56.32% |
| Russian | −48.90% | +4.68% | −23.86% | −20.30% |

Effect sizes for both sample and behavioural changes extracted from the tweet model described in Section "Tweet Modelling" between key dates. All effect sizes are relative increases in the number of tweets between the two respective dates. For the start date calculation, we drop the first two weeks of the study period. Start: start of the study period—27th January 2020. Aggression: first official US report of a mobilization of the Russian troops along the Ukrainian border—11th November 2021. War: outbreak of the war—24th February 2022. End Study: end of the study period—10th October 2022.

**Choice of language**. We analyze the choice of language more closely in the following. As we are interested in shifts between the individual languages, we look at the pairwise probability to tweet in one language over another over time. Hence, the probability reports how likely it is that a user tweets in language one (e.g. UA) over language two (e.g. EN). With three languages, this pairwise evaluation gives us a total of three different language pairs (UA over RU, UA over EN, RU over EN), where the order in which we specify each pair is irrelevant. Figure 4a visualizes how these pairwise probabilities evolved for an average user over time. For RU over EN the probability is mostly constant (82% to tweet in RU) until aggression starts, from where it continuously drops down to 58%. For UA over EN we see small increases over time (68–72%). With the mobilization of the Russian troops, we see a drop (62%), followed by a rise back to pre-aggression levels months into the war. Finally, for UA over RU we see a completely different pattern. Initially, the probability to tweet in UA is low (32%), from where it continues to rise consistently. In the weeks leading up to the war, there is a considerable speed up in this shift, resulting in a probability of 76% to tweet in UA over RU towards the end of the analysis period in October 2022.

Similarly to before, we can disentangle sample shifts from behavioural changes through statistical modelling. In summary, we fit a GAMM to model users' pairwise language probability to tweet over time, assuming a binomial distribution. As before, we include a smooth global time trend and user-specific random effects into the model. We fit such a model, for all three aforementioned language-pairs. A full description is provided in Section "Language Modelling".

Figure 4b visualizes the fitted average sample effects across all three models, i.e. the graphs depict how the average time-constant tweeting probabilities in the active user sample change over time. As we are working with coefficients of a logistic regression, changes must be interpreted with respect to changes in the odds. The figure shows that for RU over EN, initially, there is only a minor decline (−19%). However, as we approach the outbreak of the war, we can report a large drop in the odds, as users are 62% less likely to tweet in RU over EN than before, with further decreases thereafter (−29%). For UA over EN, we find a small to moderate increase until aggression (+21%) due to sample shifts, followed by a large drop until war outbreak (−52%), which is recovered in the months after (+42%). Finally, for UA over RU, there is a constant increase in the odds over time (+66%), which speeds up once aggression starts (+87% until October 2022). Table 2 details all changes. As before, changes over four-weekly

intervals are visualized in Supplementary Fig. 10. The figure shows that the sample effects for the language choice are slightly more erratic, but the major shifts take place with and after war outbreak. The alternative to Table 2, with the speed of changes is available in Supplementary Table 6, the full breakdown of all model coefficients in Supplementary Table 8.

Combining this with the results from the previous section, we can conclude that the user turnover in the first 1.5 years shifts the sample such that users are more likely to tweet in UA (than RU or EN), but not at the expense of either of the two other languages, as the sample effects for tweeting activity are (mostly) steady for both. As we approach the outbreak of the war, this drastically changes. Then, the user sample clearly shifts away from RU, as users are instead tweeting more in EN (initially) and UA (long-term). Upon further investigation (Supplementary Notes 7 and 8), we find that users tweeting in RU start leaving around November 2021 (start of aggression), with EN users joining. The former continue to leave as the war unfolds, with a few of the latter also leaving the sample again over time. This is also reflected in the increase of the UA odds over time (UA over RU consistently, UA over EN as war continues).

Figure 4c reports behavioural language changes across all three language pairs, when controlling for the user turnover. For RU over EN we see a constant decline in the odds over time (−38% to tweet in RU), which further speeds up once aggression starts (−51%). For UA over EN we see the exact opposite, as over time users are more likely to tweet in UA (+64% in odds). This change reverses with the start of aggression and the outbreak of the war (−34%), but subsequently reaches pre-aggression levels as the war unfolds. Finally, we can see a clear shift from UA to RU even early on (+129%). This switch becomes even more striking with the outbreak of the war, as users are actively changing their behaviour such that the average user is 249% more likely to tweet in UA over RU in the span of a single year. Table 2 reports all relevant changes. Supplementary Figure 10 similarly illustrates that the biggest behavioural shifts take place around the outbreak of the war, but also that there already is a constant long-term shift from UA towards RU before. Our alternative modelling strategies in Supplementary Notes 2.1 and 2.2 confirm these findings.

Connecting these language shifts with the results on tweeting activity, we find that the initial decline in EN and RU tweeting activity is not limited to monolingual users. Instead, users are actively shifting towards UA, by reducing their amount of RU and EN tweets (with a stronger shift from RU than EN respectively). Similarly, the temporary increase in EN tweeting behaviour leading up to the war can be linked to both UA and RU users. Finally and most importantly, the decline of RU and the rise of UA tweeting behaviour that manifests itself with the war is strongly driven by a major language shift (2.5 times increase) from RU to UA.

We visualize and demonstrate this substantial behavioural language shift from RU to UA in Figs. 5, 6. Figure 5 plots the language proportion of each user (UA to RU; from 0 to 1) that tweet in either language before (y-axis) and after the war (x-axis). Hence, along the straight black line through the origin we have users that do not switch language (top right UA, bottom left RU), users above the line switch to RU, below the line to UA, with users switching completely from one language to the other being located in either the top left corner (all tweets in UA to all in RU) or bottom right corner. Statistically significant ($p < 0.05$, $z > 1.96$) language shifts from before to after war outbreak are determined using a two-sided z-test with unequal variances on each user's language proportion, and are marked in the plot (the distributions were assumed to be normal but this was not formally tested). From the figure it becomes evident that there are many users that do not switch language (in both UA and RU), as well as

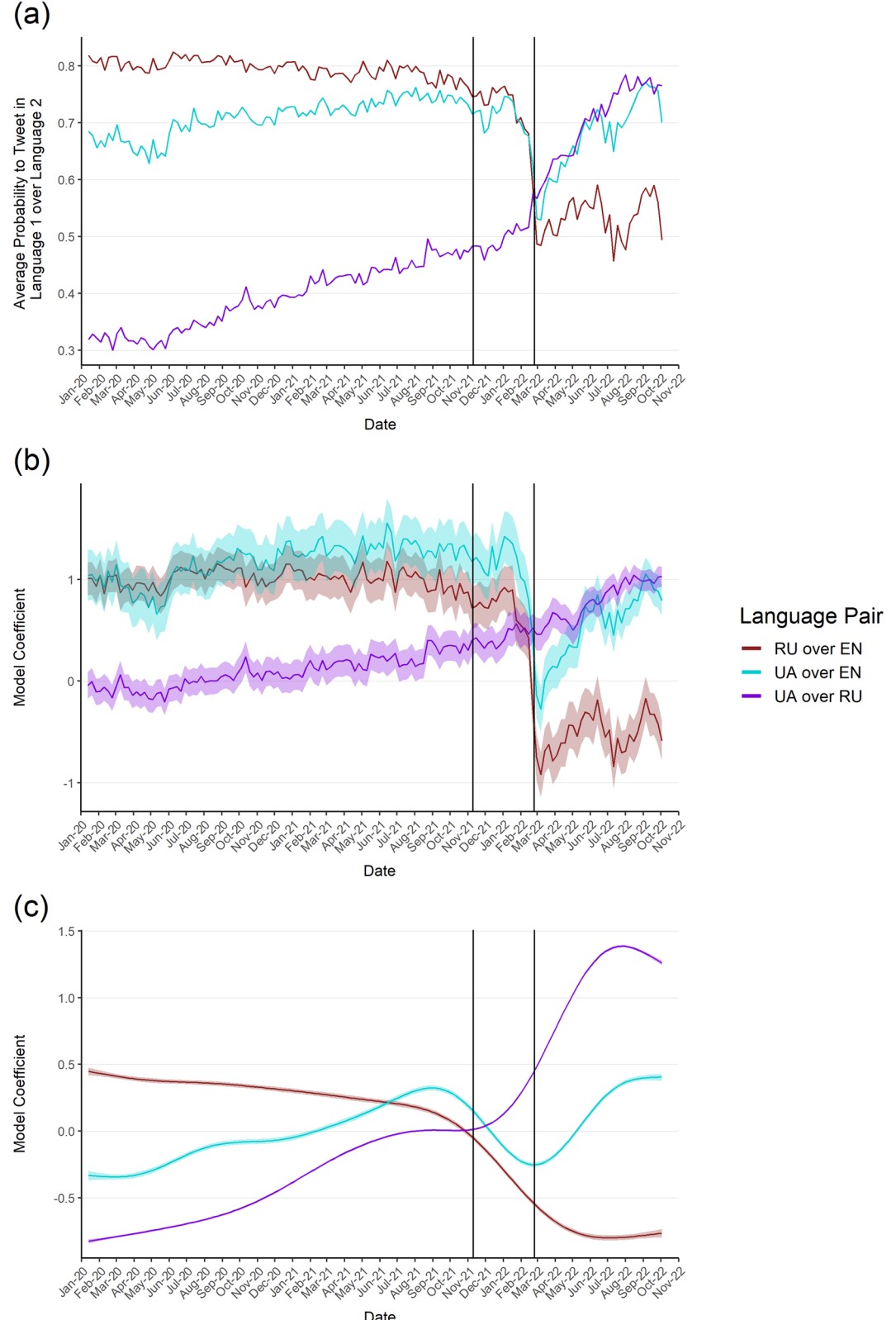

**Fig. 4 Changes in the choice of language per user (RU over EN in brown, UA over EN in turquoise, UA over RU in purple).** RU over EN in brown, UA over EN in turquoise, UA over RU in purple. (**a**) Visualizes the descriptive average probability to tweet in one language over another, (**b**) the sample effects (average random effects), (**c**) the behavioural effects (global trend). The shaded area depicts the 95% confidence interval. The first vertical line denotes the mobilization of the Russian troops along the Ukrainian border. The second line denotes the outbreak of the war.

many users clearly switching from RU to UA at various levels, whereas there are only very few switching from UA to RU.

In this sample of users who tweet in either RU or UA both before and after the outbreak of the war (3237 users), we have 1363 users who predominately tweet in RU (>80% of tweets) before the war. Of those, 839 (61.6%) tweet more in UA after the war, with 566 (41.5%) reporting a significant behavioural change ($z > 1.96$, $p < 0.05$). Out of those 850 users, 341 (25%) even switch to predominately tweeting in UA (>80% of tweets), i.e. perform a hard-switch, with 296 (21.7%) statistically significant hard-switches ($z > 1.96$, $p < 0.05$). We pick those 296 users and plot

their weekly language proportion over time in Fig. 6. Red points denote 100% of the tweets being phrased in RU, blue points denote the same in UA. From the figure, we can clearly see a substantial break and change in behaviour around the time the war breaks out (second black line), as most of the users switch from RU to UA around this mark.

On Ukrainian side, we have 1172 users who predominately tweet in UA (>80% of tweets) before the war. Of those, 471 (40.2%) tweet more in RU after the war, with only 83 (7.1%) reporting a significant behavioural change ($z > 1.96$, $p < 0.05$). More importantly, we only observe 35 (3%) hard-switches, out of

| Table 2 Language choice effect sizes between key dates | | | | |
|---|---|---|---|---|
| **Language** | **Sample Eeffects** | | | |
| | Start—Aggression | Aggression—War | War—End study | Aggression—End Study |
| UA over RU | +66.13% | +13.00% | +65.72% | +87.25% |
| UA over EN | +21.43% | −52.08% | +41.96% | −31.98% |
| RU over EN | −19.01% | −61.74% | −29.33% | −72.96% |
| | **Behavioural effects** | | | |
| UA over RU | +128.69% | +52.08% | +129.24% | +248.63% |
| UA over EN | +64.14% | −33.61% | +92.663% | +27.90% |
| RU over EN | −38.23% | −38.69% | −20.659% | −51.36% |

Effect sizes for both sample and behavioural changes extracted from the language model described in Section "Language Modelling" between key dates. All effect sizes are relative increases in the odds between the two respective dates. For the start date calculation, we drop the first two weeks of the study period. Start: start of the study period—27th January 2020. Aggression: first official US report of a mobilization of the Russian troops along the Ukrainian border—11th November 2021. War: outbreak of the war—24th February 2022. End Study: end of the study period—10th October 2022.

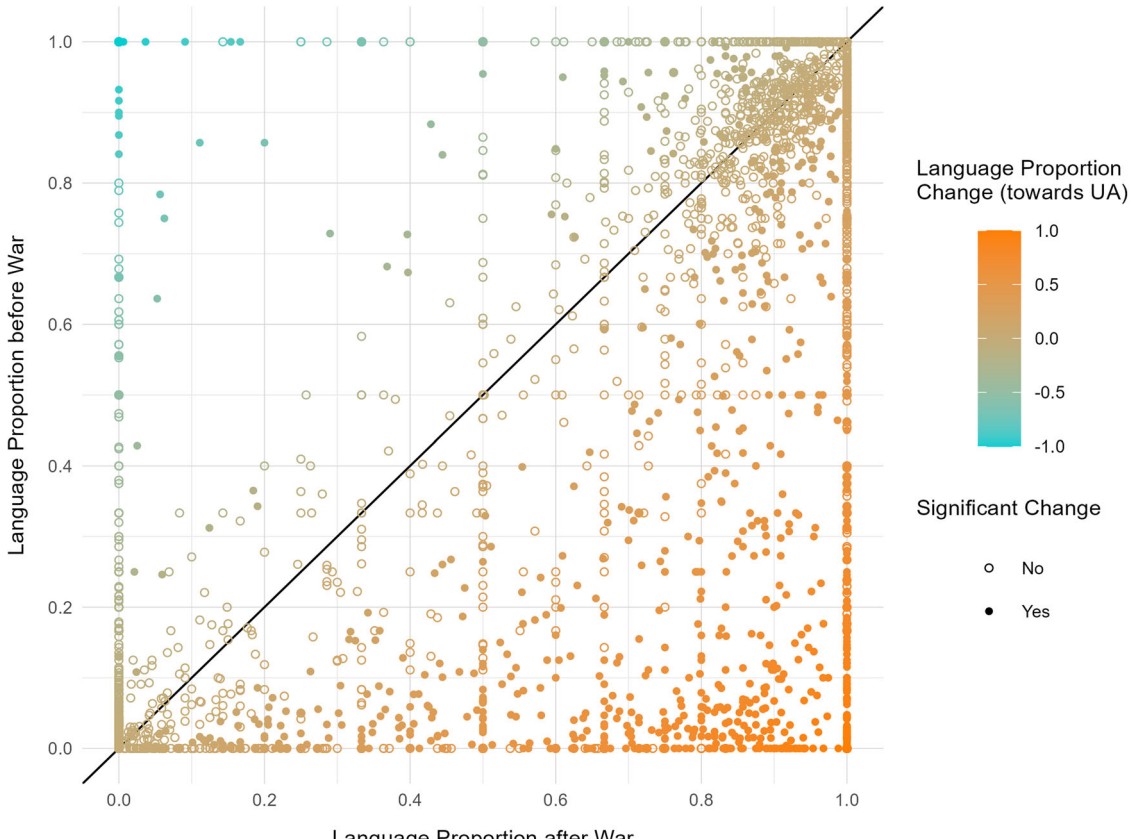

**Fig. 5 Scatterplot of users' language proportions before and after the outbreak of the war.** We are only considering users who tweet in either RU or UA (or both) before and after ($n = 3237$). The points are coloured with respect to each user's shift in language. 1 (orange) denotes a complete shift to UA, −1 (green) a complete shift to RU, 0 no shift. The straight line through the origin covers all points without a shift. Significant shifts ($z > 1.96$, $p < 0.05$) using a two-sided z-test with unequal variances on each user's language proportion are denoted through full (non-empty) points. $n = 1808$ (821 significant) shifts towards Ukrainian, $n = 818$ (106 significant) shifts towards Russian. Only RU and UA tweets of each user are considered.

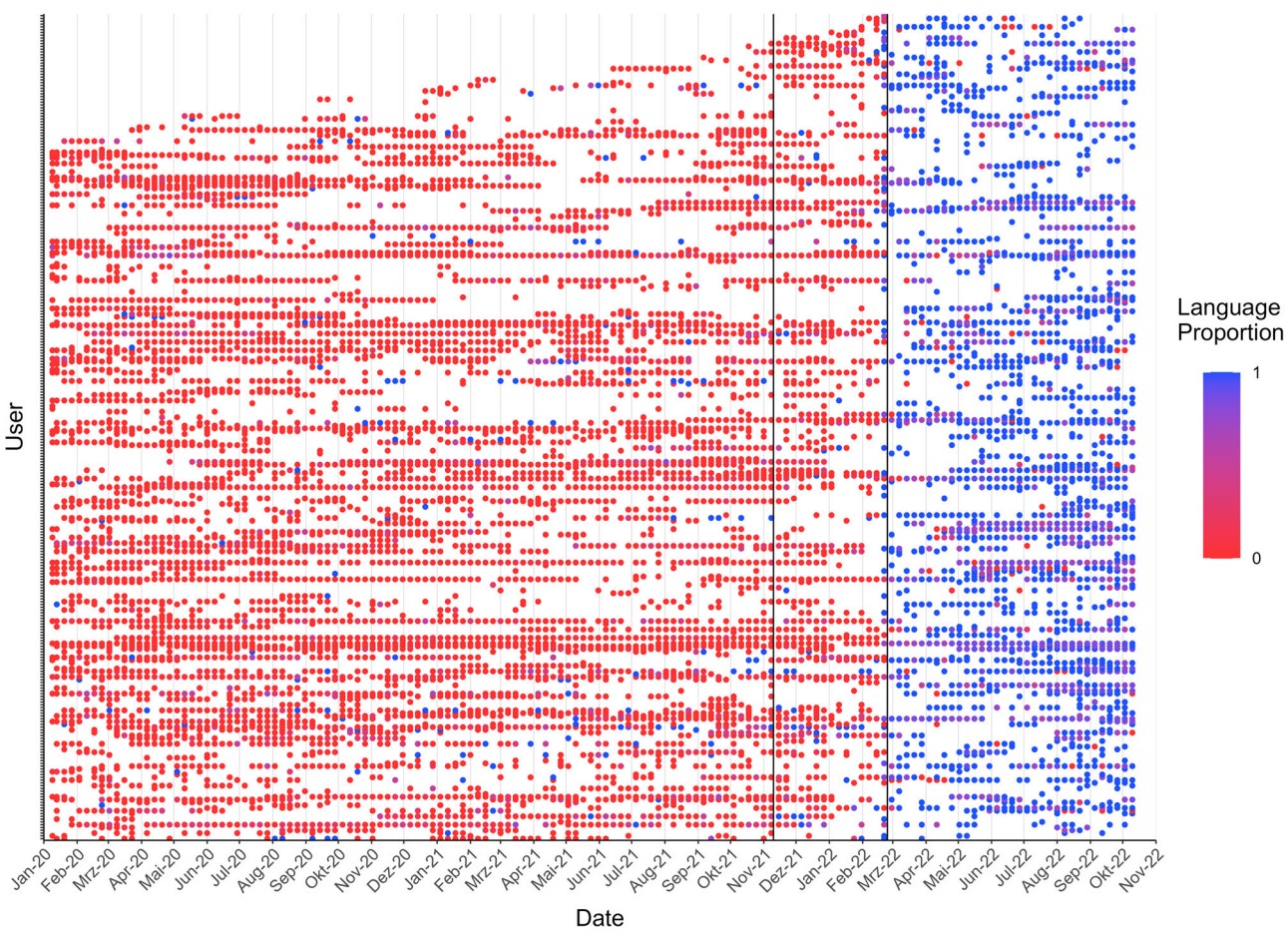

**Fig. 6 Scatterplot of users' language proportion in each week over time.** Each row (on the y-axis) denotes one of the $n = 295$ users with a statistically significant hard-switch from RU to UA. The points are coloured with respect to each user's language proportion in the respective week (145 total weeks). Blue = 100% Ukrainian, red = 0% Ukrainian (=100% Russian). Missing points indicate that a user was not tweeting in the respective week. Only RU and UA tweets of each user are considered. The first vertical line denotes the mobilization of the Russian troops along the Ukrainian border. The second line denotes the outbreak of the war.

**Table 3 Median % differences in user characteristics**

| Characteristic | No switch | Switch | Difference | P-value | $\chi^2(1)$ |
|---|---|---|---|---|---|
| Followers | 77 | 119 | **+54.54%** | 0.004 | 8.223 |
| Followings | 116 | 132 | +13.8% | 0.155 | 2.023 |
| Account age (month) | 94.15 | 105.66 | +12.22% | 0.073 | 3.196 |
| Tweet frequency | 0.79 | 1.16 | **+47.73%** | 0.021 | 5.352 |
| Likes frequency | 0.84 | 1.25 | **+48.93%** | 0.021 | 5.352 |
| # of tweets in Ukraine | 57 | 85 | **+49.12%** | 0.001 | 10.639 |
| War topic 1 | 4.0 | 6.5 | **+62.5%** | <0.001 | 22.061 |
| War topic 1 (rel.) | 0.061 | 0.063 | +4.71% | 0.801 | 0.063 |
| War topic 2 | 1 | 2 | **+100%** | 0.007 | 7.312 |
| War topic 2 (rel.) | 0.013 | 0.015 | +17.6% | 0.461 | 0.543 |

$n = 1067$ users in the no switch group, $n = 296$ users in the switch group. Column 2 reports the median of the respective user characteristic for those Russian users that do not perform a statistically significant hard-switch to Ukrainian with the outbreak of the war, column 3 for the users that do. Significant ($p < 0.05$) differences using a two-sided chi-squared are marked in bold. A description of all user attributes is provided in Section "User Characteristics".

which 20 (1.7%) are significant ($z > 1.96$, $p < 0.05$). Hence, there are only very few UA tweeting users for which we can report a significant switch towards RU after the war.

Finally, we analyze potential differences in those RU users that perform a hard-switch to UA from those that do not (see Table 3). We find that there are significant differences ($p < 0.05$) in the median in various user characteristics between the two groups using a two-sided chi-squared test (no distributional assumptions required). Users switching have more followers (+54.5%, $\chi^2(1) = 8.223$, $p = 0.004$), a higher tweet frequency (+47.7%, $\chi^2(1) = 5.352$, $p = 0.021$) as well as a higher like frequency (+48.9%, $\chi^2(1) = 5.352$, $p = 0.021$) and published more Ukraine geo-tagged tweets during the study period (+49.1%, $\chi^2(1) = 10.639$, $p = 0.001$), whereas there are only small non-significant differences in account age (+12.2%, $\chi^2(1) = 3.196$, $p = 0.07$) and followings (+13.8%, $\chi^2(1) = 2.023$, $p = 0.16$).

We also conduct a multilingual topic modelling on the tweets using BERTopic[49]. The method and its results are thoroughly described in Supplementary Notes 1. We find two topic clusters referencing the war (topic #1 and topic #3). The former is mostly related to updates regarding the situation, asking for help, and supporting the people of Ukraine, the latter covers a more political side of the overall conflict. Both topics are in total more discussed by those RU users that switch language (+62.5%, $\chi^2(1) = 22.061$, $p < 0.001$; +100%, $\chi^2(1) = 7.312$, $p = 0.007$). However, once we control for their total amount of tweets in our dataset, i.e. we compute a relative share of war related tweets for each user, the differences shrink and turn non-significant (+4.71%, $\chi^2(1) = 0.063$, $p = 0.801$; +17.6%, $\chi^2(1) = 0.543$, $p = 0.461$).

## Discussion

In our work, we collected geo-tagged tweets from Ukraine and analyzed tweeting activity and language choice before and during the Russian war in Ukraine from 9th January 2020 to 12th October 2022. Due to the nature of our longitudinal dataset and our methodological approach using a generalized additive mixed model (GAMM), we were able to disentangle and quantify shifts in the user sample, arising from user turnover, from behavioural changes of the actively tweeting users. Our GAMMs were able to handle the large sample size and take care of user's varying periods of inactivity within the study period, while at the same time allowing for a flexible non-linear but interpretable model fit.

Our analysis shows a steady long-term shift away from Russian towards Ukrainian already before the war, as the Ukrainian tweet probability rises substantially (vs. Russian; 33% to 48%). This shift is majorly driven by behavioural changes. The actively tweeting users reduce their number of Russian tweets in favour of Ukrainian over time. This is likely a conscious choice and thus shift in how the users communicate and present themselves to their online audience[26–29,31]. This finding is also in line with trends observed over a 20-year period between the 1989 and the last conducted census in 2001[36] and more recently across surveys[37], where the share of people reporting Ukrainian as their native language perpetually rose over time. Notably, with the Euromaidan protests and the subsequent Russian military intervention in 2014, this shift seems to have sped up, as citizens ethnonational identification and everyday language use is substantially shifting towards Ukrainian. This recent shift towards Ukrainian has also been identified in a small qualitative study on Facebook posts[60]. We can confirm these findings quantitatively both at-scale and in an ecologically valid setting.

We find this gradual shift to drastically speed up with the start of Russian aggression in November 2021 and the subsequent outbreak of the war. In the span of a few months, Ukrainian tweet probability rises from 48% to a remarkable 76%. While some of this increase can be explained by Russian tweeting users leaving and Ukrainian users joining (+87% in odds to tweet in Ukrainian), the major factor is a behavioural change (+249% in odds to tweet in Ukrainian), with a rise in Ukrainian (+56%) and a decrease in Russian tweeting activity (−20%). Notably, we show that out of those users predominately tweeting in Russian before the war, roughly half of them tweet more in Ukrainian after. Strikingly, around a quarter of them switch to predominately tweeting in Ukrainian, i.e., they are performing a hard-switch. It is worth noting, that we do not observe more than a handful of switches in the other direction. This shift from Russian to Ukrainian is in line with news reports and small-scale surveys outlining the war as the cause for the recent changes in language use across Ukraine[38,39]. We theorize that this is a highly politicized response. Users want to distance themselves from any

support of the war by no longer using Russian, and consciously change their self-expressed (online) identity[26–29,31], as also already to some extent reported after the Russian military intervention in 2014 both on- and offline[37,60] and confirmed in our study through the gradual shift before the war. However, with the Russian invasion, this shift seems to have sped up massively. Moreover, the distancing from supporting the war may also explain why Russian users that perform a hard-switch to Ukrainian seem to be more active on Twitter (including discussions on the war) and have a larger follower base (median of 119 vs. 77). Pressure and general interactions on social media were already reported among the main reasons for the language switch after 2014[60]. Note, that this might also (partially) explain the sample of active users shifting from Russian towards Ukrainian (sample effects).

In addition, we observe a long-term behavioural shift away from English tweeting activity up until November 2021. This could be interpreted as a reduction in talking to a broader international audience during that time[61–63], due to the fact that English is the most widely understood language on the internet by far[31,64]. However, not surprisingly, with the mobilization of the Russian troops along the Ukrainian border and specifically in the weeks leading up to the war, with a spike during outbreak, we observe a substantial shift towards English. We hypothesize users wanted to let the world know what was happening and called for aid[31], which is supported by the fact that we observe a heavy spike in English tweets assigned to the first war topic (more related to help, support and updates). While we record a large influx of English speaking users during that time (+35% in number of tweets), we can also see a substantial behavioural shift (+130%). Already active users tweet substantially more in English, independent of the language they were normally tweeting in. As the war continues to unfold, this somewhat reverses, with some of the newly joined English users leaving and behaviour reverting, although not to pre-aggression levels. With the world being more aware of the situation, and the international community supporting Ukraine in various ways[65,66], users likely have less reasons to continue tweeting in English. Instead, they return back to intra-national discussions and thus their native language(s).

**Limitations**. We recognize that while our study provides a strong foundation towards a better understanding on how the Ukrainian population reacted to the Russian invasion both on- and offline, possible limitations need to be acknowledged. The sample of users investigated here is not representative of the entire Ukrainian population. Indeed, it is skewed towards the younger and middle-aged part of the population (aged 18-49, see also Section "User Characteristics"). Additionally, we want to emphasize that geo-information is not included on most Twitter clients by default, which might further skew the sample. As on most other social media platforms, users have the option to create new accounts, which we cannot match to their prior ones. Hence, some of the behavioural effects might even be underestimated and instead accounted for as sample effects. Moreover, users might stop tweeting (with Ukrainian geo-information) for various reasons (e.g. because they fled the country). One should keep in mind that the behavioural language shifts taking place with the outbreak of the war are only demonstrated for those users who continue to tweet at and/or after the outbreak, which could potentially lead to a selection bias. Future work may analyze the content and sentiment of the tweets more closely. This could be augmented through the use of media objects attached to the tweets such as images and videos. An investigation of retweet and follower networks may reveal additional differences between those users that are shifting language to those that are not.

Naturally, any analysis can be repeated and extended to other social media platforms.

## Conclusion

In summary, our work investigated tweeting activity and language choice on Ukrainian Twitter before and during the Russian war in Ukraine through a large-scale longitudinal study. We demonstrate substantial shifts away from the Russian language to Ukrainian, which we interpret as users' conscious choice towards a more Ukrainian (online) identity. More than half of the predominately Russian-tweeting users shift towards Ukrainian, and a quarter of them even perform a hard-switch to Ukrainian, as the war breaks out. This can be seen as citizens' increasing opposition to Russia and a return to the country's linguistic roots as well as a push towards a conscious self-definition of being Ukrainian.

## Data availability

Data are available at the Open Science Framework (OSF) using https://osf.io/48sbc or with the https://doi.org/10.17605/OSF.IO/48SBC. As per Twitter developer agreement 18th April 2023, we are legally not allowed to share tweets beyond their IDs. Hence, we share our data in two ways. First, by sharing all tweet IDs needed to construct our aggregated datasets. Second, by sharing our aggregated datasets that are used for all our analyses. All of these are provided in the OSF repository with a corresponding documentation.

## Code availability

All of our code (including the aggregation scripts) is also available in the OSF repository at https://osf.io/48sbc or with the https://doi.org/10.17605/OSF.IO/48SBC. Descriptions for each script are provided there. We conducted our main analyses using R 4.1.3. In the OSF repository, we provide a session info file, which lists the version of every R package employed to conduct our analyses. We conducted the topic modelling using Python 3.10 and BERTopic 0.15.0.

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

## Acknowledgements
We would like to thank Matthias Häberle for his support in collecting the data. This work is supported by the Helmholtz Association under the joint research school "Munich School for Data Science - MUDS". This work is also supported by the European Research Council (ERC) under the European Union's Horizon 2020 research and innovation programme (grant agreement No. [ERC-2016-StG-714087], Acronym: So2Sat). The funders had no role in study design, data collection and analysis, decision to publish or preparation of the manuscript.

## Author contributions
D.R., B.I.D., P.W.T., X.X.Z. and G.K. conceived the research. P.W.T., B.I.D. and G.K. supervised the research. D.R. and G.K. designed the methodology. D.R. collected and processed the data, conducted the study and analyzed the results. D.R. created the visualizations. D.R., B.I.D., P.W.T. and G.K. wrote the first draft. All authors edited and approved the article.

## Funding

## Competing interests
The authors declare no competing interests.

## Additional information

**Peer review information** : *Communications Psychology* thanks Han-Wu-Shuang Bao and the other, anonymous, reviewer(s) for their contribution to the peer review of this work. Primary Handling Editors: Jixing Li and Antonia Eisenkoeck. A peer review file is available.

