## [Peer review file · Communications Psychology]

10th Aug 23

Dear Mr Racek,

Thank you for your patience during the peer-review process. Your manuscript titled "The Politics of Language Choice: How the Russian-Ukrainian War Influences Ukrainians' Language Use on Twitter" has now been seen by 3 reviewers, whose comments are appended below. You will see that they find your work of some potential interest. However, they have raised quite substantial concerns that must be addressed. In light of these comments, we cannot accept the manuscript for publication, but would be interested in considering a revised version that fully addresses these serious concerns.

We hope you will find the Reviewers' comments useful as you decide how to proceed. Should additional work allow you to address these criticisms, we would be happy to look at a substantially revised manuscript. If you choose to take up this option, please highlight all changes in the manuscript text file, and provide a detailed point-by-point reply to the reviewers.

Editorially, we ask you to pay particular attention to the following:

The reviewers highlight that the current study does not allow for any causal inference. We ask you to conduct the suggested regression discontinuity analysis to support any directional claims. In addition, please remove any causal claims or language from your manuscript, highlight the descriptive character of the data, and do not use directional language for correlational analyses.

All reviewers highlight issues that require further analysis of the data. This includes Reviewer 3's requests for a variation of the generalized additive mixed models analysis, but also Reviewer 2's concerns about the type of Twitter posts ("tweets", "retweets", etc) that enter the analysis, the degree to which the data are representative of typical human Twitter users, as well as the issues raised by Reviewer 1 above. In addition to these necessary analyses, we encourage you to consider the topic analysis suggested by Reviewer #2. .

Please also add a 'Limitations' section at the end of your discussion, in which you are asked to discuss issues such as a possible biased sample (e.g., geo-tagging), possible confounding factors etc.

If the revision process takes significantly longer than five months, we will be happy to reconsider your paper at a later date, provided it still presents a significant contribution to the literature at that stage.

Please use the following link to submit your revised manuscript, point-by-point response to the

Reviewers' comments with a list of your changes to the manuscript text (which should be in a separate document to any cover letter) and any completed checklist:

[link redacted]

Please do not hesitate to contact me if you have any questions or would like to discuss the required revisions further. Thank you for the opportunity to review your work.

Best regards,

Jixing Li

Jixing Li, PhD
Editorial Board Member
Communications Psychology
orcid.org/0000-0002-5210-6224

EDITORIAL POLICIES AND FORMATTING

Editorial Policy: [Policy requirements](https://www.nature.com/documents/nr-editorial-policy-checklist.pdf) (Download the link to your computer as a PDF.)

Furthermore, please align your manuscript with our format requirements, which are summarized on the following checklist:

[Communications Psychology formatting checklist](https://www.nature.com/documents/commspsychol-style-formatting-checklist-article-rr.pdf)

and also in our style and formatting guide [Communications Psychology formatting guide](https://www.nature.com/documents/commspsychol-style-formatting-guide-accept.pdf) .

* **CODE AVAILABILITY:** All Communications Psychology manuscripts must include a section titled "Code Availability" at the end of the methods section. In the event of publication, we require that the custom analysis code supporting your conclusions is made available in a publicly accessible repository; please choose a repository that provides a DOI for the code; the link to the repository and the DOI must be included in the Code Availability statement. Publication as Supplementary Information will not suffice. We ask you to prepare and upload code at this stage, to avoid delays later on in the process.

* **DATA AVAILABILITY:**

All Communications Psychology research manuscripts must include a section titled "Data Availability" at the end of the Methods section or main text (if no Methods). More information on this policy, is available at <http://www.nature.com/authors/policies/data/data-availability-statements-data-citations.pdf>.

At a minimum the Data availability statement must explain how the data can be obtained and whether there are any restrictions on data sharing. Communications Psychology strongly endorses open sharing of data. If you do make your data openly available, please include in the statement:

We recommend submitting the data to discipline-specific, community-recognized repositories, where possible and a list of recommended repositories is provided at <http://www.nature.com/sdata/policies/repositories>.

If a community resource is unavailable, data can be submitted to generalist repositories such as [figshare](https://figshare.com/) or [Dryad Digital Repository](http://datadryad.org/). Please provide a unique identifier for the data (for example a DOI or a permanent URL) in the data availability statement, if possible. If the repository does not provide identifiers, we encourage authors to supply the search terms that will return the data. For data that have been obtained from publicly available sources, please provide a URL and the specific data product name in the data availability statement. Data with a DOI should be further cited in the methods reference section.

REVIEWER EXPERTISE:

Reviewer #1 language as cultural identity, GAMM

Reviewer #2 Social media text analysis

Reviewer #3 GAMM

Reviewer #1 (Remarks to the Author):

This well-written article reports a longitudinal study of large-scale geo-tagged tweets data, finding an increasing use of the Ukrainian language and a decreasing use of the Russian language among Ukrainian Twitter users over time (2020-2022). The data show that this linguistic shift from Russian to Ukrainian was already underway before the Russian-Ukrainian War, and accelerated during the war. A remarkable strength of this study is that the authors have appropriately and successfully distinguished between user turnover effects and behavioural change effects. The data and analysis are technically sound, and the evidence is convincing. Several suggestions for further improving the clarity of this article are summarized below.

Major points:

1. While it is plausible that the war can "influence" language use (as implied in the title and abstract), readers would feel more satisfied if the authors could have empirically and quantitatively examined the causal effect of the war on language use. One possible method to consider is the regression discontinuity analysis, where the speed of change could be compared before and after the war (the cutoff) to provide evidence for its potential causal effect (beyond the gradual shift or global trend over time). Indeed, as shown in the data and stated by the authors, there has already been a long-term language shift from Russian to Ukrainian before the war, which should be accounted for more carefully if the authors wanted to make a clear causal claim.
2. The authors attributed the results largely to behavioural changes. However, a more objective description of the results (see Figures 3 & 4; Tables 1 & 2) is that the shifts in both tweeting activity and language choice are actually joint effects with *comparable* effect sizes of sample turnover and behavioural change. Although this study has its strength in disentangling behavioural effects from sample effects, it should be acknowledged that the results did not consistently indicate which one was always stronger than the other (e.g., compare Table 1 with Table 2).
3. The authors may need to elaborate on theoretical, methodological, and/or practical contributions of this study, especially on how this study can represent an advance in understanding that will influence thinking in the field. Such implications could be discussed more directly and thoroughly than simply stated as "a better understanding".

Minor points:

1. The meaning of changes in "model coefficients" (y-axis in Figures 3b, 3c, 4b, 4c) could be interpreted more clearly and explicitly for a broader audience to easily understand. Readers may be confused because "model coefficients" usually refer to regression coefficients, which are time-invariant in a model. Do these "model coefficients" actually refer to "model estimates"?
2. In Figures 3 & 4, "The shaded area depicts the 95% confidence interval of the smooth fit" (a & b) or "of the fitted effect" (c). Why are they different? Or, if these shaded areas depict the 95% CIs obtained by using the `geom_smooth()` function of the R package `ggplot2` based solely on the

time-series values (rather than *model-based 95% prediction intervals*), then they are misleading (usually overestimated on model prediction accuracy) and thus unnecessary because such CIs are not what we really want to reflect the model's prediction intervals. Indeed, the prediction intervals appropriate to display in this situation are usually wider than the 95% CIs generated by `geom_smooth()`.

3. In p. 12 "This shift from Ukrainian to Russian is in line with ..." might have been "... shift from Russian to Ukrainian ..."?

Reviewer: Dr. Han-Wu-Shuang Bao

Reviewer #2 (Remarks to the Author):

The paper investigates the choice of language (mainly Russian vs. Ukrainian) with respect to the Russian invasion to Ukraine. While the overall questions could be interesting and the approach could provide insights, there are major limitations in terms of contribution and methods of the paper.

- First and foremost, the theoretical contribution is limited; what could be the mechanisms motivating the switch from one language to another? could it be targeting audience with a certain language vs. another language over various time periods? Or is it more related to the way a user signals their identity?
- The results are at best descriptive with over presentation of raw data and lack of causal inference. What about confounding factors that could affect the language choice?
- What is the content of tweets in one language vs. another language? are the users talking about the same topics and just switch their language or they use different languages to talk about different topics? This could shed some light on the underpinning mechanisms.
- It is not clear if the authors distinguish "primary tweets", "retweets", "quotes (retweet w/ comment)", and "replies" across the paper. This could have different implication both for the sample selection and evaluating the trends. e.g when the authors filter from the API stream, did they look for original tweets, retweets, or both? This could bias the sample toward users who tweet or retweet more frequently. Similarly, it would be interesting to see how the trends vary across various types of posts (tweets, retweets, etc).
- geo-tag is not a default feature and could be turned on by the user and often for Twitter app on the phone. This makes it a highly selective sample of users. The authors need to discussion potential limitations. Additionally, selecting users from the stream API could skew the sample toward users that are more active than average.
- Who are the Twitters in this sample? could it be we are just looking at a network of Influence Operation accounts that switch from language from to another in particular periods of time to push for certain narratives? What we know about this sample characteristics? The authors could report basic users features (#tweets, #followers, ..) and more importantly the users estimated political ideology, estimated age, and quality of content shared, age of the account whether the account

belongs to an organization etc.

I hope these comments could help improve the quality of the paper.

Reviewer #3 (Remarks to the Author):

This is an interesting paper, that I read with interest and pleasure.

Major comments

I think mixed GAM is not optimal. The random intercepts $W_{\{u,l\}}$ simply shift the curve of the linear predictor along the Y-axis, resulting in parallel curves for all tweeters. This seems to me to be quite a simplification. Wouldn't a factor smooth be more appropriate: this would enable not only random intercepts, but also a random effect for the effect over time. The mgcv formula would be:

$\text{formula}(y \sim s(t) + s(t, u, bs="fs", m=1) + \dots)$

In this way, you request a random wiggly curve (the nonlinear equivalent of the combination of random intercepts and random slopes in the LMM) for each of your tweeters. You have a lot of tweeters, but the factor smooths are optimized for random-effect factors with very many levels. In this way, you can relax the assumption that all tweeters show exactly the same development over time, the only difference being the intercept. Mgcvc will generate a warning, but for this particular case, this warning can safely be ignored (Simon Wood, p.c.).

Of course, a complication is that not all of your tweeters will be active throughout the 143 weeks that you have measured their activity. My experience is that mgcv can handle situations like this gracefully. Importantly, my prediction is that once you include a time by tweeter factor smooth, the confidence intervals of your smooths for time will become wider (simply because you now allow the shape of the effect of time to vary between tweeters), and more realistic. What you report in the ms seems almost "absurdly" small credible intervals.

I had trouble understanding your "behavioral effects" and your "sample effects". This may be due to the time pressure that Communications Psychology imposes on its reviewers - I simply do not have the time to reflect on your ms as much as I would normally do. Are the behavioral effects just the first derivative (if $t_2 = t_{\{1 + 1\}}$) in $\exp(s_l(t_2) - s_l(t_1))$? I am also puzzled by your "sample effects"? Averaging over random intercepts for tweeters that are active at a given time t seems a bit weird to me. In principle, I'd like that to be part of the model - but this would require an indicator variable for every tweeter at every timestep, which is self-defeating. I think I would be helped more by a model with as response variable, for instance, the total

number of tweeters tweeting in language L at time t.

In summary, the big trends are clear and convincing; they are not going to change much (other than possibly showing more realistic, wider, credible intervals). I'd very much like to see these main trends published as soon as possible. On the other hand, I'm not sure your behavioral and sample effects are optimal. Why not leave this to a follow-up study that zooms in on the technical details? Or, alternatively, why not just provide some descriptive statistics, perhaps with lowess smoothers to bring out the main trends?

I hope the journal editors will not take these comments as a reason for rejecting your work. As I mentioned above, a simplified account of the main trend would already make for a terrific and important paper.

Please make your data available on OSF, I'd love to "play around" with your data to get a better feel for both your data and what GAMMs can do.

Minor comments

of which the last one -> the last one of which

recover almost -> recover almost all

further analysis to users, who have tweeted: delete comma (this is German comma placement, not English comma placement)

manifests with the war -> manifests itself with the war

if we are able to recover -> whether we are able to recover

Responses in this colour

Reviewer #1 (Remarks to the Author):

This well-written article reports a longitudinal study of large-scale geo-tagged tweets data, finding an increasing use of the Ukrainian language and a decreasing use of the Russian language among Ukrainian Twitter users over time (2020-2022). The data show that this linguistic shift from Russian to Ukrainian was already underway before the Russian-Ukrainian War, and accelerated during the war. A remarkable strength of this study is that the authors have appropriately and successfully distinguished between user turnover effects and behavioural change effects. The data and analysis are technically sound, and the evidence is convincing. Several suggestions for further improving the clarity of this article are summarized below.

Thank you for the positive comments about our paper and your valuable remarks, we really appreciate this. We addressed all of your suggestions in the paper and discuss them below.

Major points:

1. While it is plausible that the war can "influence" language use (as implied in the title and abstract), readers would feel more satisfied if the authors could have empirically and quantitatively examined the causal effect of the war on language use. One possible method to consider is the regression discontinuity analysis, where the speed of change could be compared before and after the war (the cutoff) to provide evidence for its potential causal effect (beyond the gradual shift or global trend over time). Indeed, as shown in the data and stated by the authors, there has already been a long-term language shift from Russian to Ukrainian before the war, which should be accounted for more carefully if the authors wanted to make a clear causal claim.

Thank you for this suggestion. We agree that a regression discontinuity (RD) design allows for interpretations based on a single parameter, before and after the intervention/treatment. However, in the given setting, we would deviate from the classic cross-sectional RD design, as time is our running variable and a specific treatment date the threshold (in our case the outbreak of the war). This is often referred to as regression discontinuity in time (RDiT) and comes with a number of issues (e.g. see Hausman and Rapson (2018)).

As discussed in Hausman and Rapson, 2018 (p.543), time-varying treatment effects violate the RDiT assumptions and are untestable. This means, a researcher must instead assume how the treatment effect evolves over time, or control units are required. Additionally, as a result, differentiating between short- and long-term effects of the treatment becomes difficult. In the given work, we show that there is a difference between the short- and longer-term behavioural effect (treatment effect). For example, with war outbreak, users are tweeting substantially more in English. But as the war continues, this behaviour somewhat reverses and the numbers are decreasing again. However, we do/did not know in advance how the treatment effect (the effect of the war) would evolve over time. In addition, we have no control units, as every user is "treated" with the war. Hence, the RDiT assumptions are violated without the possibility for a correction.

With our GAMM, we can flexibly model this without any prior assumptions on the treatment effect. (Behavioural) changes between any two points in time can be calculated using the model coefficients

(we additionally added tables for this in supplementary material S.8). Moreover, we are also able to capture the long-term (non-linear) changes over time taking place before the war.

Introducing a treatment cutoff/threshold when using higher-order polynomials to measure non-linear effects in the RDIT, might also introduce a bias into the model estimates (Hausman and Rapson (2018), p.544). We avoid this with our GAMM, as we do not use a cutoff in the model. Instead, we simply model any changes over time and measure them before and after the war.

Nonetheless, we agree that one might be interested in having additional evidence for a causal claim (which we only very carefully make). Hence, we pursue a linear RDIT design (i.e. we essentially ignore the above mentioned problems and assume a time-constant treatment effect) with the war as the treatment in supplementary material S.2.2, which we now reference to in the main text as a robustness check. We try to follow our original design using random effects as closely as possible, and show that the war indeed leads to significant changes in behaviour across all languages. The RDIT patterns closely mirror those reported in the main text, but we emphasize that the GAMM approach offers - at least to some extent - more insight in the data at hand.

We also agree that it is important to clearly communicate the speed of change over the study period, particularly comparing before and after war outbreak. While this was already partially evident from the plots in Figure 3 and Figure 4, we added two additional plots in the main paper (Figure 7 and Figure 8), which visualize behavioural and sample effects over four-weekly intervals. By using equidistant time intervals, in which we measure the changes, the reader can easily deduce the speed of the changes over time. The figures show that the biggest/fastest shifts take place shortly before, with and after the outbreak of the war. Finally, in supplementary material S.7, we also provide alternatives to Table 1 and Table 2 in the main paper, which measure the speed of the changes (instead of the total changes) between the key dates.

2. The authors attributed the results largely to behavioural changes. However, a more objective description of the results (see Figures 3 & 4; Tables 1 & 2) is that the shifts in both tweeting activity and language choice are actually joint effects with *comparable* effect sizes of sample turnover and behavioural change. Although this study has its strength in disentangling behavioural effects from sample effects, it should be acknowledged that the results did not consistently indicate which one was always stronger than the other (e.g., compare Table 1 with Table 2).

Thank you for raising this. Let us clarify this point in the following to help explain it. We have also adapted our discussion to reflect a hopefully clearer way of explaining those two effects.

The sample effects capture any changes arising from user turnover. This means any changes here are changes with respect to the overall population of active Ukrainian-based Twitter users. While this is interesting in itself, it doesn't necessarily allow us to state that language use in Ukraine is shifting, as this might simply mean that specific users (e.g. the Russian ones) stop tweeting due to the war.

The behavioural changes are much more striking. We measure these, while conditioning on the user turnover. This means any observed changes are active changes in behaviour and thus a conscious choice to adjust tweeting behaviour, here from Russian to Ukrainian. This demonstrates that the Russian

invasion indeed seems to have a strong impact on many (if not most) Ukrainian-based Twitter users. It seems that most users distance themselves from the war and show support for Ukraine through this change in identity. Hence, a comparison of the two effects is not relevant, as they measure different phenomena. We reworked our discussion to help reflect this.

3. The authors may need to elaborate on theoretical, methodological, and/or practical contributions of this study, especially on how this study can represent an advance in understanding that will influence thinking in the field. Such implications could be discussed more directly and thoroughly than simply stated as "a better understanding".

Thank you for raising this. We have substantially expanded our discussion on our methodological, and more importantly theoretical as well as practical contributions. We have also edited the introduction to reflect this.

Minor points:

1. The meaning of changes in "model coefficients" (y-axis in Figures 3b, 3c, 4b, 4c) could be interpreted more clearly and explicitly for a broader audience to easily understand. Readers may be confused because "model coefficients" usually refer to regression coefficients, which are time-invariant in a model. Do these "model coefficients" actually refer to "model estimates"?

These are the estimated model coefficients over time. The random effect (W_u in the model equation) for each user is time-constant, but different users are active at different points in time, hence the average changes over time (see Figure Figure 3b and 4b). The smooth global trend ($s(t)$) is per definition time-varying (visualized in Figure 3c and 4c). Hence, there are good reasons for using the phrase "model coefficients", even though we acknowledge that ambiguity cannot be completely avoided. We discuss our statistical models in section 4.4 and 4.5. To refer the reader to additional resources, we added two references on GAMMs in section 4.4.

2. In Figures 3 & 4, "The shaded area depicts the 95% confidence interval of the smooth fit" (a & b) or "of the fitted effect" (c). Why are they different? Or, if these shaded areas depict the 95% CIs obtained by using the `geom_smooth()` function of the R package `ggplot2` based solely on the time-series values (rather than *model-based 95% prediction intervals*), then they are misleading (usually overestimated on model prediction accuracy) and thus unnecessary because such CIs are not what we really want to reflect the model's prediction intervals. Indeed, the prediction intervals appropriate to display in this situation are usually wider than the 95% CIs generated by `geom_smooth()`.

We agree that this was slightly confusing for readers. We replaced Figures (a) and (b) by the non-smoothed versions of the plots (which were originally in our supplementary material), as the trends are still clearly evident from those figures. This means for (a) we plot the descriptive average, for (b) the mean of the random effects of the active users in each week, the latter with a 95% CI. We hope this clears up any confusion.

3. In p. 12 "This shift from Ukrainian to Russian is in line with ..." might have been "... shift from Russian to Ukrainian ..."?

Thank you for catching this. This has been corrected.

Reviewer #2 (Remarks to the Author):

The paper investigates the choice of language (mainly Russian vs. Ukrainian) with respect to the Russian invasion to Ukraine. While the overall questions could be interesting and the approach could provide insights, there are major limitations in terms of contribution and methods of the paper.

Thank you for reviewing our paper and providing feedback. We addressed your comments in the paper and report this below.

- First and foremost, the theoretical contribution is limited; what could be the mechanisms motivating the switch from one language to another? could it be targeting audience with a certain language vs. another language over various time periods? Or is it more related to the way a user signals their identity?

Thank you for raising this. We appreciate that theory has a lot to offer in terms of context for our findings. Hence, in line with Reviewer 1, we have expanded our discussion, specifically the theoretical and practical contributions with respect to why users (very likely) switch language. In our additions on the theoretical perspective, we touch upon identity and audience management, for which we have also revised our introduction. Note, that our work is exploratory and doubtlessly data driven, hence we have tried to carefully balance these additions to ensure we are not a priori hypothesizing after the results are known to us.

- The results are at best descriptive with over presentation of raw data and lack of causal inference. What about confounding factors that could affect the language choice?

We would like to respectfully disagree with this. As said before, our work is definitely data driven, but certainly more than descriptive. Our advanced statistical analyses measure and quantify tweeting activity and language choice and we are able to account for different periods and levels of activity of users throughout the study period. Conditioning on these sample effects, we show that users change their behaviour and switch language. This goes far beyond any descriptive analysis.

Our study shows substantial behavioural changes on an individual level with and after the outbreak of the war. While we do not make any explicit causal claims, we argue that the war is the main reason for this shift. We provide strong evidence in the form of our results (sudden shift with war outbreak) and support our argument with other studies highlighting similar results and rationale. In our view, it is very unlikely and unrealistic that another event happening at the exact same (or similar) point in time, i.e. a confounder, would have such a massive impact on language use, as the invasion of one's country you share a language with. Hence, this (to some extent) suggests the conclusion of causality.

- What is the content of tweets in one language vs. another language? are the users talking about the same topics and just switch their language or they use different languages to talk about different topics? This could shed some light on the underpinning mechanisms.

Thank you for this valuable suggestion. To address this, we conducted multilingual topic-modelling (which is needed due to the fact that we have three different languages) using BERTopic. We report the

full results in supplementary material S.1 and use those topics in the comparison of hard-switching users vs. non-switching users in the results. Moreover, we reference the topic modelling results in the discussion to strengthen our arguments. We emphasize however, that our primary focus was and still is on language use and we consider the topic modelling (with its limitations) as additional information.

- It is not clear if the authors distinguish "primary tweets", "retweets", "quotes (retweet w/ comment)", and "replies" across the paper. This could have different implication both for the sample selection and evaluating the trends. e.g when the authors filter from the API stream, did they look for original tweets, retweets, or both? This could bias the sample toward users who tweet or retweet more frequently. Similarly, it would be interesting to see how the trends vary across various types of posts (tweets, retweets, etc).

At page 3 of the originally submitted paper, we explain that we exclude all retweets. For clarification, we have edited the sentence to reflect that this means we include all primary tweets, quotes and replies, i.e. all tweets with original/new tweet texts. Any tweets that are too short to determine the language are excluded, as described in our methods. Note, that the sample is not biased, because we include (almost) all tweets that are posted during our study period (we show that this is indeed the case through our sensitivity analysis in section 4.2 of the main paper). Hence, all users are included irrespective of the fact that they might be more or less active. We control for this difference in activity levels in our statistical models.

- geo-tag is not a default feature and could be turned on by the user and often for Twitter app on the phone. This makes it a highly selective sample of users. The authors need to discuss potential limitations. Additionally, selecting users from the stream API could skew the sample toward users that are more active than average.

Geo-tag is indeed not a default feature. However, this is how any study using geo-information is conducted (see e.g. Hu and Wang, 2020). We have added a sentence to acknowledge this in our limitations (see discussion second to last paragraph).

Selecting users from the stream API does not skew the sample towards users that are more active. As described earlier and discussed in our paper, we were able to recover almost all tweets with Ukrainian geo-information during the study period. Hence, (almost) every user that tweeted with a Ukrainian geo-tag is included in our dataset. We hope that clears up any confusion.

- Who are the Twitters in this sample? could it be we are just looking at a network of Influence Operation accounts that switch from language from to another in particular periods of time to push for certain narratives? What we know about this sample characteristics? The authors could report basic users features (#tweets, #followers, ..) and more importantly the users estimated political ideology, estimated age, and quality of content shared, age of the account whether the account belongs to an organization etc.

Thank you for raising this. We have added a table that reports basic user features available from the API in supplementary material S.3 with a reference in section 2.1 of the main paper. We have already used those characteristics (#tweets, #followers, account age, ...) in our original paper to compare those users that switch from Russian to Ukrainian to those that do not.

Unfortunately, many of your “more advanced” suggested user characteristics are far beyond the scope of this paper (e.g. quality of content shared, which would be a study on its own given millions of tweets) or breach the Terms of Use (ToS) as per Twitter developer agreement (18th April 2023; <https://developer.twitter.com/en/developer-terms/agreement-and-policy>): *...(d) targeting, segmenting, or profiling individuals based on sensitive personal information, including their health (e.g., pregnancy), negative financial status or condition, political affiliation or beliefs, racial or ethnic origin, religious or philosophical affiliation or beliefs, sex life or sexual orientation, trade union membership, Twitter Content relating to any alleged or actual commission of a crime, or any other sensitive categories of personal information prohibited by law.*

We also want to highlight that most of the user accounts that belong to organizations are removed during our cleaning process. For example, any tweets posted by tweet schedulers such as *d/vr* are discarded, which are often used by companies or organizations. Moreover, we run an extensive spam cleaning routine (including a spam bot detection), which removes many non-human or company-run accounts from our dataset. Given the fact that we capture (almost) all users tweeting with a Ukrainian geo-tag during the study period, this should also mean that we are not just observing a network of Influence Operation accounts. The shift in language can be observed across most of our user base.

I hope these comments could help improve the quality of the paper.

Reviewer #3 (Remarks to the Author):

This is an interesting paper, that I read with interest and pleasure.

Thank you for the positive comments about our paper, we really appreciate this.

Major comments

I think mixed GAM is not optimal. The random intercepts $\$W_{\{u,l\}}\$$ simply shift the curve of the linear predictor along the Y-axis, resulting in parallel curves for all tweeters. This seems to me to be quite a simplification. Wouldn't a factor smooth be more appropriate: this would enable not only random intercepts, but also a random effect for the effect over time. The mgcv formula would be:

formula($y \sim s(t) + s(t, u, bs="fs", m=1) + \dots$)

In this way, you request a random wiggly curve (the nonlinear equivalent of the combination of random intercepts and random slopes in the LMM) for each of your tweeters. You have a lot of tweeters, but the factor smooths are optimized for random-effect factors with very many levels. In this way, you can relax the assumption that all tweeters show exactly the same development over time, the only difference being the intercept. Mgcv will generate a warning, but for this particular case, this warning can safely be ignored (Simon Wood, p.c.).

Of course, a complication is that not all of your tweeters will be active throughout the 143 weeks that you have measured their activity. My experience is that mgcv can handle situations like this gracefully. Importantly, my prediction is that once you include a time by tweeter factor smooth, the confidence intervals of your smooths for time will become wider (simply because you now allow the shape of the effect of time to vary between tweeters), and more realistic. What you report in the ms seems almost "absurdly" small credible intervals.

We initially already considered using factor smooths, but disregarded the idea due to the computational complexity this would incur. Due to your suggestion with the random wiggly curves, we revisited this for our revision. We experimented fitting factor smooths for both model specifications (tweet activity, language choice) with all 13,643 users as well as a random subsample of 1,500 users. We tried fitting the models using our preferred option $bs = "sz"$ as well as the computationally less expensive version (your suggestion) $bs = "fs"$. For this, we used a compute node offering up to 170 GB of RAM. The memory requirements were the following:

1. $bs = "sz"$
 - 13,643 users
 - Tweet activity model >170 GB RAM
 - Language choice model >170 GB RAM
 - 1,500 users
 - Tweet activity model >170 GB RAM
 - Language choice model >170 GB RAM

- 2. $bs = "fs"$
 - 13,643 users
 - Tweet activity model >170 GB RAM
 - Language choice model >170 GB RAM
 - 1,500 users
 - Tweet activity model >170 GB RAM
 - Language choice model 42 GB RAM

Hence, as a robustness check for our findings, we fitted the language choice model, using the option $bs = "fs"$, for a random subsample for 1,500 users. As we show in the supplementary material, the behavioural effects we find using this model specification are very similar to the ones we report in the main paper (with the random intercepts). We summarize all of this in supplementary material S.2.1 with a reference in the main results.

I had trouble understanding your "behavioral effects" and your "sample effects". This may be due to the time pressure that Communications Psychology imposes on its reviewers - I simply do not have the time to reflect on your ms as much as I would normally do. Are the behavioral effects just the first derivative (if $t_2 = t_{1 + 1}$) in $\exp(s_l(t_2) - s_l(t_1))$? I am also puzzled by your "sample effects"? Averaging over random intercepts for tweeters that are active at a given time t seems a bit weird to me. In principle, I'd like that to be part of the model - but this would require an indicator variable for every tweeter at every timestep, which is self-defeating. I think I would be helped more by a model with as response variable, for instance, the total number of tweeters tweeting in language L at time t .

In summary, the big trends are clear and convincing; they are not going to change much (other than possibly showing more realistic, wider, credible intervals). I'd very much like to see these main trends published as soon as possible. On the other hand, I'm not sure your behavioral and sample effects are optimal. Why not leave this to a follow-up study that zooms in on the technical details? Or, alternatively, why not just provide some descriptive statistics, perhaps with lowess smoothers to bring out the main trends?

This is correct. Our behavioural effects are simply the first derivative of s_t .

The sample effects are measured by averaging over the random intercepts for each time period t and then calculating the differences $\exp(\text{avg}(t_2) - \text{avg}(t_1))$, as described in our methods section. This is certainly not ideal, but given that factor smooths are computationally unfeasible the best option. To help the analysis, we also look at the distribution of the random effects over time through violin plots in supplementary material S.5 and S.6. We brainstormed a number of different modelling options that are able to incorporate the fact that some users have massively different tweeting intensities/numbers, but also tweet in multiple different languages in the same time period t . Ultimately, this was the best option we could come up with, given the size of the dataset and the resulting computational complexity.

I hope the journal editors will not take these comments as a reason for rejecting your work. As I mentioned above, a simplified account of the main trend would already make for a terrific and important paper.

Thank you again for your kind words regarding the paper.

Please make your data available on OSF, I'd love to "play around" with your data to get a better feel for both your data and what GAMMs can do.

Our data is already available on the OSF. We have added a more thorough description of the project structure, so you should be able to find it. If not, please do not hesitate to contact us.

Minor comments

of which the last one -> the last one of which

recover almost -> recover almost all

further analysis to users, who have tweeted: delete comma (this is German comma placement, not English comma placement)

manifests with the war -> manifests itself with the war

if we are able to recover -> whether we are able to recover

Thank you for catching those, we have edited them in the text.

10th Nov 23

Dear Mr Racek,

Your manuscript titled "The Politics of Language Choice: How the Russian-Ukrainian War Influences Ukrainians' Language Use on Social Media" has now been seen by our reviewers, whose comments appear below. In light of their advice I am delighted to say that we are happy, in principle, to publish a suitably revised version in Communications Psychology under the open access CC BY license (Creative Commons Attribution v4.0 International License).

We therefore invite you to revise your paper one last time to address the remaining concerns of our reviewers and a list of editorial requests. At the same time we ask that you edit your manuscript to comply with our format requirements and to maximise the accessibility and therefore the impact of your work.

Please note that it may still be possible for your paper to be published before the end of 2023, but in order to do this we will need you to address these points as quickly as possible so that we can move forward with your paper.

EDITORIAL REQUESTS:

SUBMISSION INFORMATION:

OPEN ACCESS:

Communications Psychology is a fully open access journal. Articles are made freely accessible on publication under a [CC BY](http://creativecommons.org/licenses/by/4.0) license (Creative Commons Attribution 4.0 International License). This license allows maximum dissemination and re-use of open access materials and is preferred by many research funding bodies.

For further information about article processing charges, open access funding, and advice and support from Nature Research, please visit <https://www.nature.com/commspsychol/article-processing-charges>

At acceptance, you will be provided with instructions for completing this CC BY license on behalf of all authors. This grants us the necessary permissions to publish your paper. Additionally, you will be asked to declare that all required third party permissions have been obtained, and to provide billing information in order to pay the article-processing charge (APC).

* **DATA AVAILABILITY:**

[link redacted]

Best regards,

Antonia Eisenkoeck

Antonia Eisenkoeck
Senior Editor

REVIEWERS' COMMENTS:

Reviewer #1 (Remarks to the Author):

The authors addressed all of my concerns adequately, which I really appreciate. The newly added supplemental analyses further corroborated their main findings. I think that the current manuscript is improved and clearer, ready for publication. I congratulate the authors on their work.

Reviewer #2 (Remarks to the Author):

The authors have address my comments and I'm happy to recommend publication.

Reviewer #3 (Remarks to the Author):

The authors have adequately addressed my concerns. It is a pity that on huge datasets, estimating generalized additive models with factor smooths is so prohibitively expensive. I think the way the authors conduct their analyses is the best that one can do at this time.